# Adaptive Visual Imitation Learning for Robotic Assisted Feeding Across Varied Bowl Configurations and Food Types

Rui Liu, Amisha Bhaskar, Pratap Tokekar

*Abstract*— In this study, we introduce a novel visual imitation network with a spatial attention module for robotic assisted feeding (RAF). The goal is to acquire (i.e., scoop) food items from a bowl. However, achieving robust and adaptive food manipulation is particularly challenging. To deal with this, we propose a framework that integrates visual perception with imitation learning to enable the robot to handle diverse scenarios during scooping. Our approach, named AVIL (adaptive visual imitation learning), exhibits adaptability and robustness across different bowl configurations in terms of material, size, and position, as well as diverse food types including granular, semi-solid, and liquid, even in the presence of distractors. We validate the effectiveness of our approach by conducting experiments on a real robot. We also compare its performance with a baseline. The results demonstrate improvement over the baseline across all scenarios, with an enhancement of up to 2.5 times in terms of a success metric. Notably, our model, trained solely on data from a transparent glass bowl containing granular cereals, showcases generalization ability when tested zero-shot on other bowl configurations with different types of food.

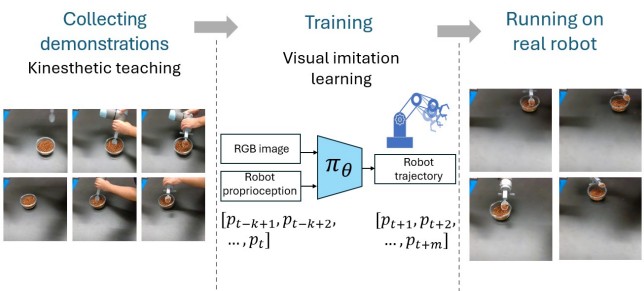

Fig. 1: Learning pipeline diagram of our approach (**AVIL**) for spoon scooping in RAF.

## I. INTRODUCTION

Inspired by human eating behavior, we pose a question: *Can robots learn to acquire food of various types from human demonstrations for assisted feeding?* In this study, we focus on spoon scooping, an essential aspect of RAF. Bhaskar et al. [1] also explored spoon scooping with a different goal of clearing the bowl. Here we aim to address the above question and effectively scoop food from a bowl. Toward this objective, we developed a novel visual imitation network to achieve adaptive scooping across varied bowl configurations and food types. The network incorporates a spatial attention module, illustrated in Figure 2, which dynamically assigns weights to different spatial locations in the input image, enabling the network to only focus on the area of interest. We name our approach **AVIL**, which stands for Adaptive Visual Imitation Learning.

To validate our approach, we tested it on a real robot and compared its performance with a baseline represented by a handcrafted scooping motion. The results demonstrate that our model outperforms the baseline across all varied bowl configurations and food types. Notably, we trained the model solely with data collected from granular cereals in a transparent glass bowl. Despite this, the model exhibited effectiveness when tested zero-shot with plastic bowls of different sizes, as well as with other food types such as semi-solid jelly and liquid water. Moreover, we assessed the model's robustness by subjecting it to tests with distractors

All authors are from the Department of Computer Science, University of Maryland, College Park, MD 20742, USA. Email: {ruiliu, amishab, tokekar}@umd.edu

on the table. Even in the presence of distractors, the model maintained its performance, showcasing its robustness and resilience.

## II. APPROACH

Employing visual imitation learning, we build a robust framework that integrates imitation learning to directly map input observations, including RGB images and robot proprioception (joint positions), to corresponding robot control actions. The learned policy is adaptive to variations in bowl position, size, material, and food types. Figure 1 illustrates our learning pipeline. This process involves collecting human expert demonstrations, training the model using our visual imitation network, and then deploying the learned policy on a real robot.

### A. Preliminaries

*1) Observation and Action Space:* In our visuomotor policy learning system of spoon scooping for RAF, the input observation space $\mathcal{O}_t = (\mathcal{I}_t, p_t)$, where $\mathcal{I}_t \in \mathbb{R}^{3 \times H \times W}$ represents the RGB images captured from a static environment camera, and $p_t \in \mathbb{R}^6$ denotes the robot proprioception, representing the 6D joint positions. The state $s_{t-k:t} = (\mathcal{I}_t, p_{t-k+1}, p_{t-k+2}, \ldots, p_t)$ involves the RGB image of the current timestep $t$ and a sequence of last $k$ steps of joint positions. The action $a_{t:t+m} = (p_{t+1}, p_{t+2}, \ldots, p_{t+m}) \in \mathbb{R}^{m \times 6}$ involves the predicted joint positions for the next $m$ steps.

*2) Visuomotor Policy Learning:* We formulate the policy learning problem as a supervised learning task with behavior cloning, aiming to learn a parameterized policy $\pi_\theta$ with the following objective function:

$$\theta = \arg\min_\theta \mathbb{E}_{(s_{t-k:t}, a^*_{t:t+m}) \sim \mathcal{D}}[\mathcal{L}(\pi_\theta(s_{t-k:t}), a^*_{t:t+m})], \quad (1)$$

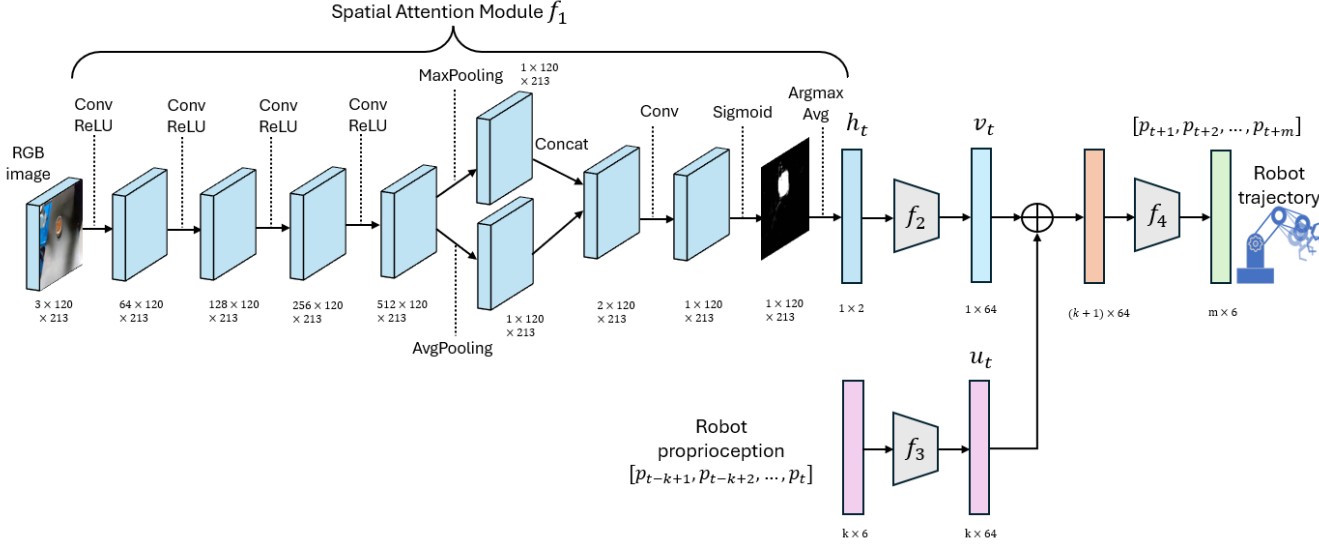

Fig. 2: Proposed visual imitation network.

where $\theta$ is the parameters of the policy, $\pi_\theta(s_{t-k:t})$ is the predicted actions for the state $s_{t-k:t}$, and $a^*_{t:t+m}$ is the expert action. $\mathcal{L}$ is the mean squared error loss.

To initiate the visuomotor policy learning process, we first collect a set of demonstrations as the training data. Our dataset $\mathcal{D} = \{(s_{t-k:t}, a^*_{t:t+m})\}^N_{i=1}$ consists of $N$ robot trajectories obtained through kinesthetic teaching, where each trajectory $i$ comprises pairs of states and actions.

After completing the preliminaries and formulating the visuomotor policy learning problem, we proceed to tackle the problem and learn the policy using our visual imitation network.

### B. Visual Imitation Policy Network

In this section, we introduce the main component of our paper, the visual imitation policy network, designed to learn visuomotor policies for scooping various food items. The network is crucial for mapping visual observations and robot proprioception into robot actions. The network's objective is to minimize the discrepancy between the predicted robot actions $\pi_\theta(\cdot)$ and the demonstrated actions throughout the learning process.

To enhance learning from the historical data, our policy network takes the RGB image of current timestep $t$ and a sequence of last $k$ steps of robot proprioception (joint positions), $(\mathcal{I}_t, p_{t-k+1}, p_{t-k+2}, \ldots, p_t)$, as input. The network's outputs consist of the predicted joint positions for the subsequent $m$ steps, $a_{t:t+m} = \pi_\theta(s_{t-k:t})$, which is a practical innovation. This approach, unlike solely predicting a single step in previous work [2], offers the advantage of mitigating error accumulation over time and providing a more detailed view on future trajectories. During inference on a real robot, we adopt the initial step from the predicted $m$ future steps for execution, drawing inspiration from Model Predictive Control (MPC). Then we observe and update the state $s_{t-k+1:t+1} = (\mathcal{I}_{t+1}, p_{t-k+2}, \ldots, p_t, p_{t+1})$.

The architecture of our visual imitation network, outlined in Figure 2, incorporates several key components: a spatial attention module, a visual attention embedding module, a robot proprioception embedding module, and a control action module. The output layer generates predicted joint positions for the next $m$ steps, which are subsequently compared with expert actions during the training process. Notably, our proposed network incorporates a spatial attention module. This module enables the model to focus on crucial areas within the image, facilitating improved adaptation for different bowl configurations and food types for our RAF system.

*a) Spatial attention module:* This module dynamically weights different spatial locations in the input image and allows the robot to selectively attend to specific areas of interest, such as the food source or the targeted bowl in our case, contributing to the development of a more refined and context-aware RAF system.

We depict the architecture of the spatial attention module in Figure 2. The module comprises four convolution layers for image feature extraction, each followed by a ReLU activation [3]. Each layer employs a filter size of 3, stride 1, and padding 1. In this way, we can maintain the spatial dimensions of the feature map for an input image while increasing its depth.

Drawing inspiration from the convolutional block attention module in [4], we apply channel-wise max pooling and average pooling to the intermediate feature map, resulting in two one-dimensional feature maps. Then we concatenate these two feature maps along the channel, and pass through a convolution layer and a sigmoid layer, we obtain the spatial attention map. Notably, different from the approach in [4], we introduce an auxiliary binary cross-entropy loss to guide the learning of the spatial attention module.

*b) Visual attention embedding module:* A linear projection layer that projects the visual attention feature obtained from the spatial attention module into higher dimensions.

*c) Robot proprioception embedding module:* A linear projection layer that projects the robot proprioception (joint angles) into higher dimensions.

*d) Control action module:* A Multi-Layer Perceptron (MLP) with several fully connected layers for mapping features to robot actions.

As mentioned before, we can decompose our visual imitation network $\pi_\theta$ into four modules. Mathematically, $\pi_\theta = (f_1, f_2, f_3, f_4)$, where $f_1$ is the spatial attention module with parameters $\theta_1$, $f_2$ is the visual attention embedding module with parameters $\theta_2$, $f_3$ is the robot proprioception embedding module with parameters $\theta_3$, and $f_4$ is the control action module with parameters $\theta_4$.

$$h_t = f_1(\mathcal{I}_t; \theta_1), \tag{2}$$
$$v_t = f_2(h_t; \theta_2), \tag{3}$$
$$u_t = f_3(p_{t-k+1}, p_{t-k+2}, \ldots, p_t; \theta_3), \tag{4}$$
$$a_{t:t+m} = f_4(v_t, u_t; \theta_4). \tag{5}$$

For our proposed visual imitation network, given the RGB image $\mathcal{I}_t$, it initially undergoes the spatial attention module $f_1$, producing a one-channel attention map. By performing an argmax operation on this map, we derive the coordinates representing the area of interest in the image. A subsequent averaging operation allows us to determine the 2D visual attention feature $h_t = (x_t, y_t)$, essentially representing the centroid of this region. Then we project $h_t$ into higher 64 dimensions via the linear embedding $f_2$. Simultaneously, the last $k$ steps of 6D robot joint positions are also projected into higher 64 dimensions via the linear embedding $f_3$. By projecting data into a higher-dimensional space, our model gains more capacity to represent complex functions. This increased expressiveness allows the model to capture non-linear relationships that might not be discernible in lower dimensions. We concatenate the resulting visual feature vector $v_t$ and embedded robot proprioception vector $u_t$ and proceed through the control action module $f_4$, ultimately produce the robot actions $a_{t:t+m}$.

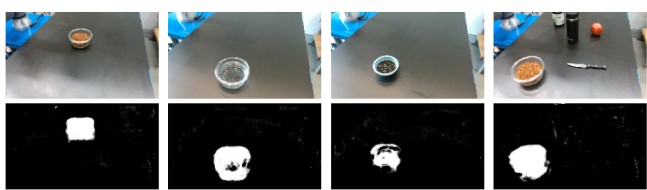

Fig. 3: Qualitative results of images with various bowl configurations, positions, food types, and with distractors on the table along with their corresponding spatial attention maps.

We present some qualitative results that highlight the effectiveness of our spatial attention module and its robustness against distractors after training. For training details, refer to [5]. We display images depicting various bowl configurations, positions, and food types along with their corresponding spatial attention maps in Figure 3. Specifically, we show a transparent glass bowl containing cereals,

followed by its contents shifting to water at a different position. Subsequently, we depict a plastic bowl containing jelly. Additionally, we introduce distractor items such as a water bottle, an apple, a jelly jar, and a knife onto the table to simulate a realistic kitchen environment. The spatial attention module effectively focuses on the bowl area, as evidenced by the corresponding attention map. This allows the visual imitation network to accurately scoop the desired food despite the presence of distractors. Notably, our spatial attention module is much smaller, requires less computation, and is faster to train compared to pretrained object detectors like RetinaNet [6], Yolo [7], and Mask-RCNN [8], yet achieves effective performance in extracting visual features for our RAF application.

## III. Experiments

After training the visual imitation network, we test it on a real robot and compare its performance with a baseline. We test across varied bowl configurations, positions, and food types. For the detailed experimental setup, please refer to [5].

### A. Baseline

For the baseline, we first utilize RetinaNet [6] to detect the bowl given an RGB image. Upon obtaining the bounding box, we calculate the centroid of the bowl. Subsequently, we map this position to the robot coordinate and direct the robot to move to that position with a specific height and orientation. Then, we adjust the wrist 2 joint of the robot arm to rotate by $-0.6$ radians to initiate the scooping action.

During testing on different bowl positions, the baseline maintains a consistent end-effector height and orientation to reach the bowl centroid. Additionally, the rotation of the wrist 2 joint remains fixed at $-0.6$ radians. We do not customize different baseline settings for varied bowl configurations and food types.

### B. Experimental Results

For both **AVIL** and basline, we test across varied bowl configurations, food types, and different positions. For each bowl configuration, food type, and position, we conduct five trials of scooping attempts. We use a success metric criterion. We assign a numerical value of 1 to successfully scooping food items from a bowl without spillage. We consider instances where some spillage occurs as partial success and assign a numerical value of 0.7. And we assign a numerical value of 0 to failure cases.

*1) **AVIL and Baseline Performance Comparison:*** We evaluate the performance of our approach **AVIL** and compare it with the baseline. To provide comprehensive comparisons, we average the success metrics over different aspects: when comparing success metrics across varied bowl configurations, we average over food types and bowl positions; likewise, when comparing success metrics across different food types, we average over bowl configurations and positions; and when comparing success metrics across bowl positions, we average over bowl configurations and food types.

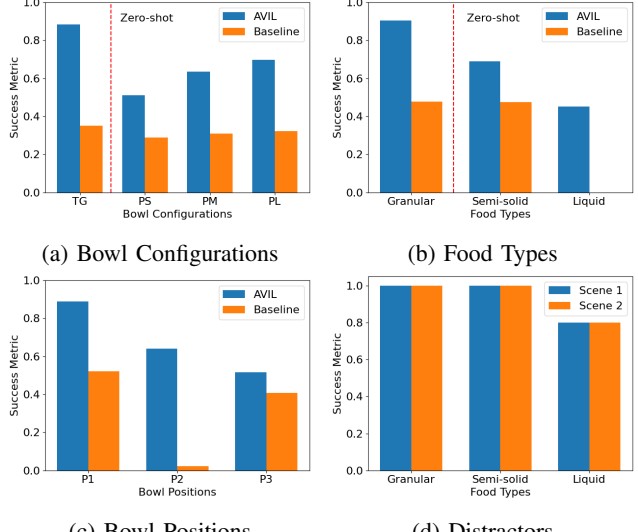

(a) Bowl Configurations

(b) Food Types

(c) Bowl Positions

(d) Distractors

Fig. 4: **Experimental results**. (a-c) Performance comparison between **AVIL** and baseline across varied bowl configurations, food types and bowl positions. We present on the right side of the dashed red line in (a-b) conditions tested zero-shot. (d) **AVIL** performance with and without distractors. Scene 1 represents without distractors and Scene 2 represents with distractors.

*a) Across varied bowl configurations:* In Figure 4a, we illustrate the comparison results across varied bowl configurations, encompassing different bowl materials and sizes. The results demonstrate that **AVIL** consistently outperforms the baseline. Specifically, the success metric is 2.5, 1.8, 2.1, and 2.2 times higher compared to that of the baseline for TG, PS, PM, and PL bowls, respectively. The TG bowl achieves the highest success metric, as we collected the training data using this bowl. Additionally, among plastic bowls, the PS bowl has the lowest success metric, likely due to its smaller size, which increases the likelihood of spillage.

*b) Across varied food types:* In Figure 4b, we present the comparison results across varied food types, including granular, semi-solid and liquid. Notably, the baseline struggles particularly with scooping liquid water due to its inherent property of flowing away. The handcrafted scooping motion employed by the baseline, involving rotation of the wrist 2 joint of the robot arm, proves insufficient in handling this challenge. In contrast, our approach adapts to such challenges by learning from human demonstrations, enabling effective scooping even with liquids. For granular and semi-solid food types, the success metric of **AVIL** is 1.7 and 1.3 times higher compared to that of the baseline, respectively.

*c) Across varied bowl positions:* In Figure 4c, we show the comparison results across varied bowl positions. For a fair comparison, **AVIL** and baseline have the same robot starting configuration for each trail of scooping attempts. The baseline faces difficulties with position P2, as it only directs the robot to move to the centroid of the bowl without planning a path in between. Consequently, the spoon collides

with the bowl and fails to enter it during scooping attempts. However, our approach overcomes this limitation by learning from human demonstrations and effectively navigating the spoon into the bowl. For P1 and P3, the success metric of **AVIL** is 1.9 and 1.5 times higher compared to that of the baseline, respectively.

*2) Zero-shot Generalization:* Our data collection process exclusively involved the transparent glass bowl containing granular cereals. However, during testing, we evaluated **AVIL** on plastic bowls of various sizes and containing different food types such as liquid and semi-solid. Remarkably, **AVIL** demonstrates effective performance across these varied plastic bowls, as depicted in Figure 4a on the right side of the dashed red line, with higher success metrics compared to the baseline for PS, PM, and PL bowls.

In Figure 4b, positioned to the right of the dashed red line, when testing on semi-solid and liquid food types, VILA also exhibits effective performance. Specifically, the success metric for semi-solid is higher than that for liquid, which aligns with the expectation that liquid is more likely to flow away. Meanwhile, the baseline method proves ineffective for scooping liquid due to its inadequate motion.

*3) Robustness Against Distractors:* In the above Figure 3, we demonstrate the effectiveness of our spatial attention module and its robustness against distractors. Here in Figure 4d, we present a performance comparison of **AVIL** with and without distractors. We denote Scene 1 as the condition without distractors and Scene 2 as the condition with distractors. We conduct tests on VILA using the TG bowl at position P1 with various food types, performing five trials of scooping attempts for each food type. As depicted in Figure 4d, both Scene 1 and Scene 2 exhibit identical performance. This suggests that distractors do not influence the performance of **AVIL**, verifying the robustness of **AVIL** against distractors.

## IV. CONCLUSIONS

We introduce a novel visual imitation network with a spatial attention module for spoon scooping in RAF. Our approach, named AVIL (adaptive visual imitation learning), demonstrates adaptability and robustness, effectively handling varied bowl configurations in terms of material, size, and position, as well as diverse food types including granular, semi-solid, and liquid, even in the presence of distractors. This overcomes the drawbacks of previous work with limited adaptability to different container configurations and food types. To validate our approach, we conduct comprehensive experiments on a real robot and compare its performance with a baseline. The results demonstrate clear improvement over baseline across all variations, with an enhancement of up to 2.5 times in terms of a success metric, validating the efficacy of our model.

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
