# OpenReview forum: "Adaptive Visual Imitation Learning for Robotic Assisted Feeding Across Varied Bowl Configurations and Food Types"
_IEEE.org/2024/ICRA/Workshop/CookingRobot — CookingRobot2024 Poster_

### Official Review · Reviewer_zkZg · 2024-04-15
**Review of "Adaptive Visual Imitation Learning for Robotic Assisted Feeding Across Varied Bowl Configurations and Food Types"**

**Rating:** 8
**Confidence:** 4

**Review:**

This paper proposes a visual imitation network with a spatial attention module along a spoon scooping task. It thoroughly validates the proposed method compared with baseline results with varying bowl configurations, food types, and with distractors on the physical robot.

Comments on Paper:
* It addresses an important problem in cooking, tackling tasks in varying visual scenes and in the presence of other objects, utilising spatial attention to improve robustness.
*  The experiment is well-executed with various unseen food types and containers on a physical robot.
* It is interesting to use spatial attention to improve robustness to different scenes and distractors. However, it is unclear what the novelty is compared to existing methods using spatial attention for imitation learning.
* This work compares the proposed method of adaptive imitation learning against baseline, which is a handcrafted motion. As the contribution of the proposed method is to use spatial attention in imitation learning, I would be interested to see the comparison of the method using the same imitation learning method with and without spatial attention.

Comments on Video:
* Nice video showing scooping various food items from varying containers using baseline and proposed methods.

---

### Official Review · Reviewer_Qufk · 2024-04-15
**Review of "Adaptive Visual Imitation Learning for Robotic Assisted Feeding Across Varied Bowl Configurations and Food Types"**

**Rating:** 7
**Confidence:** 4

**Review:**

The paper proposes an approach for robotic assisted feeding (RAF) with visual imitation learning. The learned policy shows generalizability for bowl size and food owing to the spatial attention network in the proposed model.

Questions
- To what extent the leaned model can be transferred to foods with different dynamics?

Comments on the video
- Showing evaluation with different setting (position, shape of bowl, and food) is effective for illustrating the robustness of learned model.
- Presenting the scenes of data collection will make the video clear about the experiment procedures.